# Hypertension Is Associated with Increased Risk of Diabetic Lung

**DOI:** 10.3390/ijerph17207513

**Published:** 2020-10-15

**Authors:** Jihyun Lee, Donghwan Kwon, Youngjang Lee, Inchan Jung, Daesung Hyun, Hunju Lee, Yeon-Soon Ahn

**Affiliations:** 1Department of Medicine, Wonju College of Medicine, Yonsei University, 20 Ilsan-ro, Wonju 26426, Korea; deoventree@gmail.com (J.L.); kdhwan1996@yonsei.ac.kr (D.K.); dudwkd1220@gmail.com (Y.L.); inchan1996@yonsei.ac.kr (I.J.); 2Department of Preventive Medicine, Wonju College of Medicine, Yonsei University, 20 Ilsan-ro, Wonju 26426, Korea; hyonds@gmail.com (D.H.); hjlee5371@gmail.com (H.L.); 3Department of Preventive Medicine and Genomic Cohort Institute, Yonsei Wonju College of Medicine, Yonsei University, Wonju 26426, Korea

**Keywords:** diabetic lung, hypertension, diabetes, impaired fasting glucose, pulmonary function

## Abstract

Lung function is often impaired in diabetic patients, especially in a restrictive pattern, which has recently been described as the diabetic lung. Since hypertension (HTN) is common in diabetic patients, our study investigated whether HTN acts as an aggravating factor in diabetic lung. Within the cross-sectional study from the 6th Korean National Health and Nutrition Examination Survey (KNHANES), fasting plasma glucose (FPG), blood pressure (BP), pulmonary function, and laboratory data were examined in 4644 subjects aged between 40 and 79 years. A multivariate regression model was used to investigate the relationship between BP, FPG, and pulmonary function. Lung function was significantly reduced in the HTN (*p* = 0.001), impaired fasting glucose (IFG) (*p* < 0.001), and diabetes mellitus (DM) (*p* < 0.001) groups. Next, a multivariate logistic regression model was used to derive the odds ratio (OR) of reduced lung function based on the presence of IFG, DM, and HTN. The OR of reduced forced vital capacity (FVCp < 80%) was 3.30 (*p* < 0.001) in the HTN-DM group and 2.30 (*p* < 0.001) in the normal BP-DM group, when compared with the normal BP-normal FPG group. The combination of HTN and DM had the strongest negative effect on FVC. The results presented in this study indicate that diabetes and hypertension have a synergistic association with impaired lung function.

## 1. Introduction

Over the past few decades, many researchers have revealed that diabetes impairs pulmonary function in a restrictive pattern—known as the diabetic lung [1,2,3,4]. Chronic hyperglycemia produces proinflammatory glycosylated proteins, which are deposited in small vessels and connective tissues [5]. Glycation of collagen and elastin of lung tissue cause stiffening of the parenchyma [6]. Systemic inflammation is being pointed as a link between diabetes and lung dysfunction. Abnormal regulation of inflammatory mechanisms could cause exaggerated inflammatory responses in the lung, resulting in impaired lung function [7].

Moreover, since the Framingham study found an inverse relationship between forced vital capacity (FVC) and cardiovascular diseases in 1983 [8], several studies consistently showed an association between hypertension and impaired pulmonary function [9,10]. Some studies even revealed that the coexistence of hypertension and reduced lung function was associated with a higher mortality rate [11,12]. Although the pathologic mechanism is not entirely investigated, chronic systemic and pulmonary inflammation is being regarded as the main mechanism. 

Given the fact that hypertension is common among patients with diabetes and is a strong risk factor for microvascular complications [13], there is a need to evaluate whether blood pressure (BP) impacts on the reduction of lung function in diabetes. However, it is still unknown whether hypertension acts as an aggravating factor in diabetic lung. Therefore, we designed this cross-sectional study to prove the following: (1) if diabetes or hypertension is correlated with reduced lung function at a nationwide level and, if it is correlated, (2) to what extent does hypertension affect lung function in diabetic and nondiabetic populations.

## 2. Materials and Methods

### 2.1. Study Population and KNHANES 

The Korean Centers for Disease Control and Prevention (KCDC) conducted the 6th Korean National Health and Nutrition Examination Survey (KNHANES) from 2014 to 2015 [14]. This is a nationwide cross-sectional survey with a stratified multistage clustered probability sampling design to select a representative sample of civilian, noninstitutionalized Korean adults aged at least 18 years. The KNHANES comprises mainly two parts. First, a comprehensive self-reported questionnaire on medical history and health-related behaviors. Second, clinical examination and biochemical laboratory measurements, including BP assessment, pulmonary function test, liver function test, and blood glucose level. For biochemical measurements, the participants underwent blood sampling after 8 h of overnight fasting. The KCDC and related academic societies managed the external and internal quality control programs for all steps (including survey administration, data collection, laboratory analysis, and data processing). The laboratory data quality control program monitored the laboratory performances to ensure that all analytical values meet acceptable standards of precision and accuracy. In addition, 30 expert committees comprising over 120 experts technically supported KNHANES regarding quality assurance and control of the survey and the selection of individual survey items [14].

Of the 14,930 individuals who participated in KNHANES 2014–2015, 8626 individuals aged ≥40 years were included in the present study, because spirometry was measured in participants aged 40 years or older. The study exclusion criteria applied were: (1) did not undergo a pulmonary function test (*n* = 9075); (2) missing BP, fasting plasma glucose (FPG) level, and HbA1c values (*n* = 636); and (3) missing health-related questionnaire replies (*n* = 575). Finally, a total of 4644 participants (2120 males and 2524 females) were included in the analysis (Figure 1). All voluntarily engaged participants provided written informed consent prior to their enrollment. The participants’ records, except survey date and home region, were anonymized prior to analysis.

### 2.2. Pulmonary Function Test 

Pulmonary function was measured by clinical technicians using a dry rolling seal spirometer (model 2130; Sensor-Medics, Yorba Linda, CA, USA). Forced expiratory volume in 1 s (FEV1) and FVC measurements were performed according to the American Thoracic Society/European Respiratory Society guidelines and National Health and Nutrition Examination Survey (NHANES) method. Lung function measurements primarily included the FVC, FEV1, forced expiratory volume in 6 s (FEV6), forced expiratory flow between 25% and 75% of the FVC (FEV25–75%), and peak expiratory flow (PEF). Four criteria were applied to the spirometer data: (1) at least three acceptable spirometry curves for 6 s or more, (2) <150 mL inter-measurement variability in FVC and FEV1, (3) ≤10% inter-measurement variability in PEF, and (4) extra volume ≤5% of FVC (150 mL). Predicted spirometry values were calculated from the Korean reference equations, based on representative samples of the Korean population [15]. We used FVC, FEV1, the FEV1 to FVC ratio (FEV1/FVC), and the percentage of predicted values for FEV1 (%, FEV1p) and FVC (%, FVCp) to assess pulmonary function. A decline in the FVCp < 80% was regarded as a restrictive pattern of poor lung function [16].

### 2.3. Diabetes and Impaired Fasting Glucose

Plasma glucose levels were measured after an overnight fast using the Hitachi Automatic Analyzer 7600 (Hitachi, Tokyo, Japan). HbA1c was measured by high-performance liquid chromatography (HPLC) assay (HLC-723G7; Tosoh, Tokyo, Japan).

Individuals were classified as having diabetes if any of the following criteria were met, adapted from the 2010 American Diabetes Association (ADA) criteria [17]: FPG level ≥126 mg/dL (7.0 mmol/L) and HbA1c ≥6.5%. Having impaired fasting glucose (IFG) was defined as an FPG level of 100 mg/dL (5.6 mmol/L)–125 mg/dL or HbA1c 5.7–6.4%, according to the 2010 ADA criteria.

### 2.4. Hypertension and Prehypertension

BP was measured by four nurses in charge of BP measurement in the Special Investigation Department of the KCDC. It was measured manually using a standard mercury sphygmomanometer (Baumanometer Wall Unit 33(0850); WA Baum Co., Inc., Copiague, NY, USA) with a cuff of an appropriate size. BP was measured thrice (1 min intervals) with the subject in the sitting position after resting for at least 10 min, with the right arm supported at the level of the heart. The mean of the second and third measurements was used in the analysis. 

The 2018 European Society of Cardiology/European Society of Hypertension (ESC/ESH) Guidelines for the management of arterial hypertension was used to classify the BP status [18]. Hypertension was defined as systolic BP (SBP) ≥140 mmHg or diastolic BP (DBP) ≥90 mmHg. Prehypertension was defined as SBP 120–139 or DBP 80–89 mmHg. Normal BP was defined as SBP <120 mmHg and DBP <80 mmHg. If SBP and DBP fell into different categories, the category associated with the higher of the two pressures was applied. 

### 2.5. Height, Weight, Waist Circumference, and Body Mass Index 

Height, weight, and waist circumference were measured using standard techniques and equipment. Height was measured to the nearest 0.1 cm with a portable stadiometer (Seca 220; Seca, Hamburg, Germany). 

Weight was measured with a calibrated balance beam scale (Giant-150N; HANA, Seoul, Korea) with the participants wearing a light gown and no shoes. 

Waist circumference was measured in the standing position at the level of the mid-point between the lower rib margin and the iliac crest with full expiration. All parameters were measured by well-trained staff with regular quality assurance check-ups. 

Body mass index (BMI) was calculated by dividing the weight in kilograms by the height in meters squared (kg/m^2^). 

### 2.6. Handgrip, Smoking History, Alcohol Consumption, and Physical Activity

The handgrip strength of each hand was measured thrice using a digital grip strength dynamometer (TKK 5401; Takei Scientific Instruments Co., Ltd., Tokyo, Japan). The measurements of handgrip strength were presented as an average of the maximal values of the left and right hands. 

We defined a smoker as a person who had smoked more than five packs of cigarettes (100 cigarettes) in their lives [19]. They were further classified as a current smoker or former smoker depending on the smoking status at the time of the interview. Everyone else was defined as a nonsmoker. 

Harmful alcohol use is often defined as the consumption of <60 g of pure alcohol per drink per day for men and >40 g for women [20]. When converted into the units of drink, it corresponds to ≥ 7 glasses of Soju (Korean beverage that usually contains 20% of alcohol by volume) per occasion for men and ≥5 glasses of Soju for women. Therefore, we defined heavy alcohol consumption status as drinking seven or more glasses of alcohol on one or more occasions per week in men and five or more glasses on one or more occasions per week in women. One glass is equivalent to a glass of alcohol without distinguishing between beers, Soju, or any other liquor, as KNHANES did not distinguish them. 

Participants who engaged in aerobic physical activity were defined as those who engaged in moderate-intensity physical activity for at least 2 h and 30 min per week or high-intensity physical activity for 1 h and 15 min. (1 min of high-intensity workout = 2 min of moderate-intensity workout). Further details of the measurements from KNHANES VI can be found on their website (http://www.knhanes.cdc.go.kr).

### 2.7. Statistical Analysis

SPSS software version 25.0 for Windows (IBM Inc., Armonk, NY, USA) was used for all analyses. Differences in the demographic and anthropometric characteristics according to the hypertensive and diabetic status were compared using the one-way analysis of variance (ANOVA), followed by the post-hoc Bonferroni correction.

A multivariate linear regression analysis was performed after adjusting for several variables to confirm the relationships between hypertension-pulmonary function and diabetes-pulmonary function. Models were initially run without adjusting covariates and then repeated after adjusting for age, sex, BMI, waist circumference, lifetime smoking, alcohol consumption, aerobic physical activity engagement, and handgrip strength.

Then, we stratified the study population into six groups depending on the presence of diabetes, prediabetes, and hypertension: normal FPG-normal BP, normal FPG-hypertension, IFG-normal BP, IFG-hypertension, diabetes-normal BP, and diabetes-hypertension. A multivariate logistic regression model was used to compare the degree of lung function impairment in each group. *p*-value < 0.05 was considered statistically significant.

### 2.8. Ethical Statement

All subjects provided their informed consent to participate in this study. The study was conducted in accordance with the Declaration of Helsinki, and the protocol was approved by the Ethics Committee of the Yonsei University Wonju Severance Christian Hospital, Wonju, Republic of Korea (CR320353).

## 3. Results

The characteristics of the 4644 participants stratified by the presence of diabetes and prediabetes are summarized in Table 1. Continuous and categorical variables were expressed as mean ± standard deviation and *n* (%), respectively. In total, 666 participants presented with diabetes, and 2175 participants presented with prediabetes. The highest mean age in the diabetes group was 61.71 years. The diabetes group comprised more male subjects than the normal BP group. SBP was significantly lower in the nondiabetes group than in the diabetes group. The highest mean SBP was 125.21 mmHg in the diabetes group, and the lowest mean SBP was 117.23 mmHg in the nondiabetes group. However, DBP showed no statistical tendency. The mean FPG and HbA1c were 90.00 mg/dL and 5.35% in the nondiabetes group, 100.95 mg/dL and 5.80% in the prediabetes group, and 145.49 mg/dL and 7.32% in the diabetes group, respectively. The lowest mean FVCp was 89.11% in the diabetes group, and the highest mean FVCp was 95.83% in the nondiabetes group. The lowest mean FEV1/FVC was 0.76 in the diabetes group, and the highest mean FEV1/FVC was 0.78 in the nondiabetes group. There was a significant difference among the three groups in FVCp, FEV1p, and the FEV1/FVC ratio.

We performed a linear regression analysis to investigate the associations between the glycemic parameters, BP, and pulmonary function (Table 2). An inverse relationship was observed between the glycemic status and both FVCp and FEV1p. However, the FEV1/FVC ratio did not show statistical relevance with the glycemic status. After adjusting for potential confounding factors associated with pulmonary function, including age, sex, BMI, waist circumference, smoking status, heavy alcohol consumption status, aerobic physical activity engagement, and handgrip strength, FVCp and FEV1p still showed an inverse relationship. The regression coefficient (β) of FVCp was −3.639 (*p* < 0.001) in the diabetes group and −1.496 (*p* < 0.001) in the prediabetes group. In the hypertension group, FVCp was reduced compared with the normal BP group. The regression coefficient (β) was −1.528 (*p* = 0.001) after adjusting for confounding factors. However, this inverse relationship was not observed in the prehypertension group. The regression coefficient (β) of FVCp in the prehypertension group was -0.320 (*p* = 0.395). Additionally, neither FEV1p nor FEV1/FVC were significantly correlated with hypertension and prehypertension.

To further investigate the effects of hypertension on the diabetic lung, a multivariate logistic regression analysis was performed (Table 3). We divided the population into six subgroups based on the presence of hypertension, diabetes, hypertension, and prediabetes. The odds ratio (OR) of restrictive lung disease (FVCp < 80%) was highest in the subjects who had both hypertension and diabetes (OR 6.47; 95% CI: 4.22–9.93). Even after adjustment for age, sex, BMI, waist circumference, smoking history, alcohol consumption, aerobic physical activity, and handgrip strength, it had a significant effect on restrictive lung disease (OR: 3.30; 95% CI 2.09–5.22). The OR of restrictive lung disease was 2.30 (95% CI: 1.66–3.18) and 1.93 (95% CI: 1.34–2.77) in the diabetes group without hypertension and prediabetes group with hypertension, respectively.

## 4. Discussion

In this cross-sectional study, we found that hypertension was strongly associated with reduced pulmonary function, particularly in FVCp. Both the prediabetes and diabetes groups showed reduced pulmonary function, especially in FVCp and FEV1. These relationships remained significant even after adjustment for potential confounding factors. On comparison, prehypertension did not show a statistical relationship with lung function. Moreover, from a statistical point of view, the risk of reduced pulmonary function was greater when diabetes and hypertension coexisted. The OR of restrictive lung disease (FVCp < 80%) was 3.30 in the hypertension and diabetes group (95% CI: 2.09–5.22) and 2.30 in the normal BP and diabetes group (95% CI: 1.66–3.18). This finding may indicate that poor BP control in diabetes increases the risk of pulmonary dysfunction and disease acceleration. The OR of restrictive lung disease was significantly lower in females (OR:0.21; 95% CI: 0.14–0.31); moreover, on investigating the risk of restrictive lung disease after stratifying the subjects by sex, the results remained statistically significant. The OR of restrictive lung disease in males was 2.87 (95% CI: 1.56–5.25) in the hypertension and diabetes group. In females, the OR was 2.07 (95% CI: 1.44–4.68) (data not shown in any table).

As the number of patients with diabetes and hypertension are ineluctable, our findings show a considerable clinical impact on population levels. Although the contribution of hyperglycemia and hypertension to the total OR is minor, in those diagnosed, clinical signs and symptoms do not tend to be isolated clinical findings. Therefore, an extensive assessment of pulmonary function is warranted and should be recommended for hyperglycemic populations with poor BP control. When we designed this study, we planned to evaluate the effect of “prehypertension” and “prediabetes”, because diabetes and hypertension develop gradually over many years, and the recent paradigm of medicine focuses on prevention rather than treatment. However, the results showed that prehypertension alone could not be regarded as a risk factor for lung function deterioration.

The Framingham Heart Study, conducted in Massachusetts, United States, 1983 (*n* = 3254) [8], first reported that diabetes was associated with a lower level of FVC. Numerous cross-sectional and longitudinal studies conducted after 1983 supported this result. A recent case-control study conducted in Egypt in 2013 (*n* = 100) [21] and a cross-sectional study conducted in the Republic of Korea in 2013 (*n* = 9233) [22] proved that IFG and not only diabetes was associated with lower FVC and FEV1/FVC. A case-control study conducted in Germany in 2018 (*n* = 255) [23] found that breathlessness in combination with restrictive lung disease was significantly increased in patients with prediabetes and type 2 diabetes. Vascular inflammation induced by hyperglycemia is regarded as a key factor in lung function deterioration, and the association between hyperglycemia and vascular inflammation is well-established [24].

Moreover, cardiovascular complications of hypertension substantially overlap with hyperglycemia [25,26,27,28]. Diabetes and hypertension are comparable as contributors to vascular damage. Since the endothelial function is impaired in hypertension, vasodilation is impaired, and the vascular tone increases. Eventually, it falls into the proinflammatory and prothrombic state, which causes vascular remodeling. The lungs consist of numerous capillaries, so it could be hypothesized that hypertension affects lung tissue in an inflammatory manner. It also has been hypothesized that renin-angiotensin-aldosterone system activation in hypertension, as well as diabetes, contributes to accelerated vascular damage [29,30]. Moreover, the upregulation of an inflammatory reaction in hypertensive vascular tissue plays a significant role in cardiovascular complications [31,32,33]. A case-control study conducted in Finland and Israel in 2008 (*n* = 700) demonstrated that C-reactive protein (CRP) concentrations increased with higher HbA1c levels [34] and higher BP [35]. Quynh N D et al. even insisted in their study that anti-inflammatory drugs could potentially be used to treat hypertension [36]. In a study conducted in Iceland in 2009 (*n* = 758) [37], hypertension had a negative effect on FVC, while hypertension and high CRP were independent and additive determinants of FEV1. The results of the Women’s Health and Aging Studies (WHAS study I and II, *n* = 1172) [38] showed that the combined elevation of interleukin-6 (IL-6) and high-sensitivity-CRP (hs-CRP) was associated with the lowest pulmonary function levels in women of advanced age. Previous data compared favorably with our findings, which were related to the association between hypertension-lung function and diabetes-lung function. In addition, by analyzing how the pulmonary function results differ when BP and FPG levels were considered, respectively and simultaneously, our study reinforced the existing results from a new perspective. This study revealed that diabetic subjects with poorly controlled BP were more likely to develop restrictive lung disease than those with controlled BP.

Potential effect modification between the variables can be assessed as a departure from additive effects. Relative excess risk due to interaction (RERI) is considered the standard measure for interaction on the additive scale [39]. If it is modified to reflect our use of OR, RERI (OR _diabetes and hypertension_ − OR _non diabetes and hypertension_ − OR _diabetes and normal BP_ + 1) is 0.55. This indicates that diabetes and hypertension have a synergistic association beyond an additive association.

This is the first study to describe the association between diabetic lung and hypertension. The main strengths of this study are the large sample size and the nationwide population-based setting. All survey measurements and data collection were standardized by a trustworthy organization. The use of standardized BP and FPG-level examinations also facilitated the comparisons. Our study also included the HbA1c level in addition to the FPG level when stratifying the diabetic population. There were also some limitations to the study that should be acknowledged. First, pulmonary function was only performed in subjects aged 40 years or older, so it was impossible to evaluate the effect of hypertension in young generations. Second, inflammatory reactants such as the erythrocyte sedimentation rate (ESR) and C-reactive protein (CRP) levels were not considered. Therefore, the next study could be done to compare the degree of deterioration rate of lung function in hypertension-hyperglycemic populations with and without elevated inflammatory reactant concentrations. Finally, the cross-sectional design limited our ability to determine the causal effect between exposures and outcomes. Therefore, further prospective studies are needed to evaluate the temporal sequence and causality between hypertension and diabetic lung.

## 5. Conclusions

We speculate that hypertension in combination with diabetes and not diabetes itself might be associated with reduced lung function in the general adult population. Our findings support the need for pulmonary function evaluation, treatment, and follow-up on those presenting with diabetes and hypertension. However, it must be considered that, in our analyses, hypertension might merely be an indicator of uncontrolled diabetes and that only the combination with very severe hypertension is associated with lung function impairment.

## Figures and Tables

**Figure 1 ijerph-17-07513-f001:**
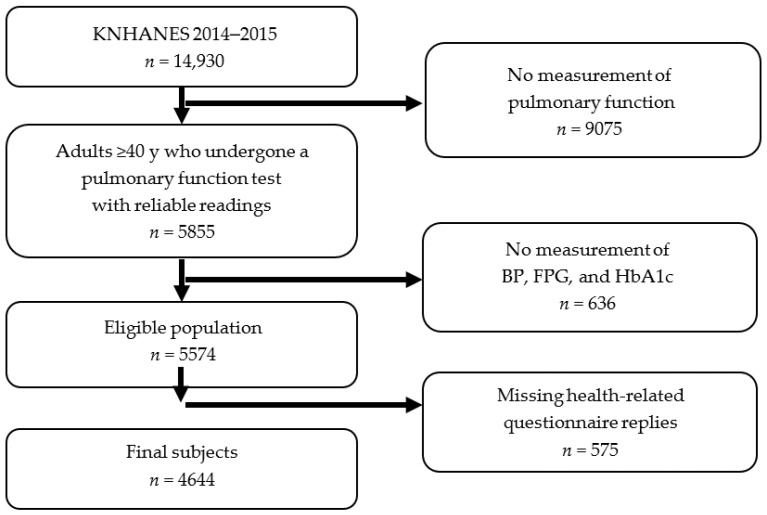
Flow diagram showing the selection of study participants. Abbreviations: KNHANES, Korean National Health and Nutrition Examination Survey; BP, blood pressure; FPG, fasting plasma glucose; HbA1c, Hemoglobin A1c

**Table 1 ijerph-17-07513-t001:** The characteristics of the participants stratified by the glycemic status.

Characteristics Mean ± STD or *n* (%)	Normal FPG; A (*n* = 1803)	IFG; B (*n* = 2175)	DM; C (*n* = 666)	*p*-Value * (Post Hoc)
Sex				
Male	681 (32.1)	1060 (50.0)	379 (17.9)	<0.001 (A<B<C)
Female	1122 (44.5)	1115 (44.1)	287 (11.4)	<0.001B>C)
Age	54.01 ± 10.21	58.74 ± 10.25	61.71 ± 9.75	<0.001 (A<B<C)
FPG (mg/dL)	90.00 ± 5.66	100.95 ± 9.30	145.49 ± 40.75	<0.001
HbA1c (%)	5.35 ± 0.21	5.80 ± 0.30	7.32 ± 1.16	<0.001
BMI (kg/m^2^)	23.26 ± 2.78	24.56 ± 3.06	25.15 ± 3.19	<0.001
Waist circumference (cm)	79.98 ± 8.42	84.56 ± 8.71	87.48 ± 8.61	<0.001 (A<B<C)
Lifetime smoking status				
Nonsmoker	1169 (42.7)	1233 (45.1)	334 (12.2)	<0.001 (A>B>C)
Former smoker	374 (33.3)	547 (48.7)	202 (18.0)	<0.001 (A<B<C)
Current smoker	260 (32.7)	395 (49.7)	140 (17.6)	<0.001 (A<B=C)
Regular physical activity				
Yes	907 (50.3)	1030 (47.4)	309 (46.4))	0.098
No	896 (40.4)	1145 (45.9)	357 (13.8)	
Heavy drinking	310 (17.2)	436 (20.0)	146 (21.5)	0.022 (A<B=C)
Handgrip strength	30.90 ± 9.35	31.81 ± 9.57	31.75 ± 9.34	0.007 (A<B=C)
Blood pressure				
Systolic BP (mmHg)	117.23 ±16.30	122.16 ± 16.71	125.21 ± 15.19	<0.001 (A<B<C)
Diastolic BP (mmHg)	75.84 ± 9.91	76.65 ± 9.99	75.42 ± 10.37	0.005 (A=C<B)
Blood pressure status				
Normal BP	937 (52.0)	883 (40.6)	213 (32.0)	<0.001 (A>B>C)
Prehypertension	604 (33.5)	874 (40.2)	312 (46.8)	<0.001 (A<B<C)
Hypertension	262 (14.5)	418 (19.2)	141 (21.2)	<0.001 (A<B=C)
Pulmonary function pattern				
Restrictive (FVCp < 80%)	113 (6.3)	266 (12.3)	150 (22.1)	<0.001 (A<B<C)
Obstructive (FEV1/FVC < 0.7)	196 (10.9)	356 (16.5)	142 (20.9)	<0.001 (A<B<C)

* *p*-value was obtained from one-way analysis of variance, followed by the post-hoc Bonferroni correction. Abbreviations: FPG, fasting plasma glucose; IFG, impaired fasting glucose; DM, diabetes mellitus; BP, blood pressure; FVC, forced vital capacity; FEV1, forced expiratory volume in 1 s; and BMI, body mass index.

**Table 2 ijerph-17-07513-t002:** Association between blood pressure, glycemic status, and pulmonary function using a multivariate linear regression analysis.

	Crude Model		Adjusted *	
	β	*p*-Value	β	*p*-Value
	FVCp (%)			
Normal FPG	1 (Ref)		1 (Ref)	
IFG	−3.020	<0.001	−1.496	<0.001
DM	−6.342	<0.001	−3.639	<0.001
Normal BP	1 (Ref)		1 (Ref)	
Prehypertension	−1.515	<0.001	−0.320	0.395
Hypertension	−2.832	<0.001	−1.528	0.001
	FEV1p (%)			
Normal FPG	1 (Ref)		1 (Ref)	
IFG	−2.086	<0.001	−1.628	<0.001
DM	−4.605	<0.001	−3.422	<0.001
Normal BP	1 (Ref)		1 (Ref)	
Prehypertension	−0.445	0.311	−0.204	0.647
Hypertension	−0.678	0.225	−0.621	0.274
	FEV1/FVC			
Normal FPG	1 (Ref)		1 (Ref)	
IFG	−0.016	<0.001	0.001	0.683
DM	−0.024	<0.001	0.004	0.171
Normal BP	1 (Ref)		1 (Ref)	
Prehypertension	−0.013	<0.001	0.002	0.435
Hypertension	−0.010	0.001	0.007	0.008

* Adjusted for age, sex, BMI, waist circumference, smoking history, alcohol consumption, aerobic physical activity, and handgrip strength. Abbreviation: β: regression coefficient.

**Table 3 ijerph-17-07513-t003:** Multivariate logistic regression model and odds ratio of restrictive lung disease (FVCp < 80%) according to hypertension, glycemic status, and other covariates.

	Crude	Adjusted *
Subgroups		
Normal BP–Normal FPG (*n* = 1541)	1 (Ref)	1 (Ref)
Normal BP–IFG (*n* = 1757)	2.21 (1.70–2.87)	1.49 (1.13–1.97)
Normal BP–DM (*n* = 525)	4.33 (3.20–5.87)	2.30 (1.66–3.18)
HTN–Normal FPG (*n* = 262)	1.94 (1.24–3.06)	1.45 (0.91–2.33)
HTN–IFG (*n* = 418)	3.23 (2.31–4.54)	1.93 (1.34–2.77)
HTN–DM (*n* = 141)	6.47 (4.22–9.93)	3.30 (2.09–5.22)
Age		1.02 (1.00–1.03)
Sex		
Male		1 (Ref)
Female		0.21 (0.14–0.31)
BMI		1.11 (1.05–1.18)
Waist circumference		1.02 (1.00–1.05)
Smoking history		
Nonsmoker		1 (Ref)
Former smoker		0.83 (0.61–1.13)
Current smoker		0.99 (0.71–1.38)
Aerobic physical activity engagement		0.77 (0.64–0.94)
Drinking history		
Not a heavy drinker		1 (Ref)
Heavy drinker		0.66 (0.73–1.23)
Handgrip strength		0.92 (0.91–0.94)

* Adjusted for age, sex, BMI, waist circumference, smoking history, alcohol consumption, aerobic physical activity, and handgrip strength.

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
