# Peer review of "Hypertension Is Associated with Increased Risk of Diabetic Lung"

_ijerph, 2020, doi:10.3390/ijerph17207513_

Round 1

Reviewer 1 Report

The Authors designed the cross-sectional study to prove if diabetes and hypertension are correlated with reduced lung function in a nationwide level and  how much did hypertension affect lung function in diabetes and nondiabetes population. As the Authors themselves say the main strengths of this study are the large sample size and the nation-wide population-based setting. Authors found that hypertension was strongly associated with reduced pulmonary function, particularly in FVCp. This study revealed that diabetic subjects with poorly controlled BP were more likely to develop restrictive lung disease than those with controlled BP.

SUGGESTIONS

It would be appropriate to insert the References in subsections 2.3 and 2.5

"Diabetes group comprised more male subjects than did the normal BP group": this statement is not reflected in table 1.

"The highest mean SBP was 125.21 mmHg in diabetes group, and the lowest mean SBP was 117.23 mmHg in nondiabetes group": it would be appropriate to insert in table 1 the mean + SD of the BP and FVCp values divided into the 3 groups of glycemic state.

line 190: there is a writing error: the regression coefficient of FVCp (-3.649) does not correspond to that in table 2 (-3.639)

References must be standardized: for example, international abbreviations of biomedical journals, initial and final page etc.

Author Response

Dear reviewer.

 First, we appreciate the time and effort that you have dedicated to providing your valuable feedback on our manuscript. We are grateful for your insightful comments on our paper.

 We have been able to incorporate changes to reflect most of the suggestions provided by you. We have highlighted the changes within the manuscript. Here is a point-by-point response to your comments and concerns.

  1. It would be appropriate to insert the References in subsections 2.3 and 2.5

Unfortunately we could not understand exactly what you have pointed out. I hope you give us the detailed information about which reference to be supplemented with so that we can understand. Subsection 2.3 have the reference (2010 American Diabetes Association (ADA) criteria) and subsection 2.5 doesn't need certain reference because it is explaining about a measurement method of height, weight and waist circumference.

  1. "Diabetes group comprised more male subjects than did the normal BP group": this statement is not reflected in table 1.

We changed the table 1 to clearly show the percentage.

  1. "The highest mean SBP was 125.21 mmHg in diabetes group, and the lowest mean SBP was 117.23 mmHg in nondiabetes group": it would be appropriate to insert in table 1 the mean + SD of the BP and FVCp values divided into the 3 groups of glycemic state.

We inserted rows in table 1 to show the mean BP value.

  1. line 190: there is a writing error: the regression coefficient of FVCp (-3.649) does not correspond to that in table 2 (-3.639)

We edited the number. Thank you so much for your note.

  1. References must be standardized: for example, international abbreviations of biomedical journals, initial and final page etc.

We edited the references. 

 We look forward to hearing from you in due time regarding our submission and to respond to any further questions and comments you may have. Again, thank you for your review.

Sincerely,

J Lee, D Kwon, Y Lee, I Jung, D Hyun, H Lee and Y Ahn

Reviewer 2 Report

Thank you, I am honoured you allow me to review and share some considarations and suggestions on your work and manuscript.

This is a concise piece of scientific research. I have no major comments on the content. The abstract and the body of text are well written. 

The (scientific and clinical) enthusiasm of the authors can be sensed and basically the text reads well with a good flow for the reader.

This is not to criticize the scientific content of the authors, but it would help if the text receives an English grammatical tune-up. This concerns minor grammatical and textual issues like the correct use of verbs and the use of single/plural which already become apparent in the Abstract and continue throughout the Body of text. The paper would therefore greatly merit if reviewed by a native English speaker. 

Some questions/suggestions:

Abstract. Well written and clear outline. Please check for language (this is important as readers will scan the Abstract first prior to further reading).

Well written and clear means and method section.

Paragraph 2.1. This section does well describe the subject selection procedures and includes a clear Figure on the process. How was the self-reported medical status quality controlled? Also, I suggest a line of text is attributed to how the authors dealt with the pitfall of selection bias? 

Paragrapgh 2.4. Blood pressure is a major endpoint. Were blood pressure devices standardized and validated? I assume blood pressure was measured in the non-dominant upper arm according to RR?

Paragraph 2.5. There are many measurements done in this study. How (and how often) were quality assurance procedures performed or was this a continuous process? Were there a priori set quality control requirements for the measurements?

Paragraph 2.6. Line 135. I am not an expert in the field of Korean liquors. What is Soju?

Results section. Good presentation of the Results. I am always most happy to notice the p-values have been corrected for multiple testing (Bonferroni's correction, Table 2). This makes a solid impression, in particular as this in part is an exploratory study. All abbreviations are systematically and well explained in the footnotes of the Tables; this also contributes to the good readability.

The discussion is interesting and shows the clinical relevance of the study. A matter of semantics is the following (line 239): 'The clinical relevance for the individual might be small, but it cannot be neglected on the 239 population level. Therefore, assessment of pulmonary function should be recommended for 240 hyperglycemic population with poor blood pressure control'. This sentence may - after some puzzling - be modified with the last part of the first sentence brought forward. The study warrants this approach in the conclusion: starting the sentence with 'The clinical relevance for the individual might be small,....etc.' is a too humble approach for the work done. This is only a matter of wording. An approach could for example be: 'As the numbers in the elderly population with diabetic lung and hypertension are sizeable, our findings show a considerable clinical impact at population levels. Also, although the contribution of diabetic lung and hypertension to total RR is minor, in  those diagnosed these clinical signs and symptoms do not tend to be isolated clinical findings. Therefore, extensive assessment of pulmonary function is warranted and should be recommended for hyperglycemic population with poor blood pressure control' This is just a suggestion also in the light of the important clinical message (line 241 - 245): 'When we designed this study, we planned to evaluate the effect of ‘prehypertension’ and ‘prediabetes’, because blood pressure and glucose level elevation develops gradually over many years. And recent paradigm of medicine especially focuses prevention rather than treatment' The following sentence is very important (line 245): 'But the result showed us that prehypertension alone could not be regarded as a risk factor in lung function deterioration'.

Line 298. The concluding sentences are phrased in a very careful and almost humble matter. In itself this is a very good approach from a philosphical perspective. I am not a reviewer that likes marketing blurps like 'Take home message' in scientific papers. However, clinicians tend to like some practical guidance. This Reviewer thinks the authors may, based on their well executed science, add a stronger clinical relevant statement in the concluding sentences to make the findings more palatable for the treating physicians, for example 'Our findings support the need for extensive clinical evaluation, treatment and follow-up in those presenting with diabetic lung and hypertension'.

Hope this helps.

Author Response

Dear reviewer. 

 First, we appreciate the time and effort that you have dedicated to providing your valuable feedback on our manuscript. We are grateful for your insightful comments on our paper.

 We have been able to incorporate changes to reflect most of the suggestions provided by you. We have highlighted the changes within the manuscript. Here is a point-by-point response to your comments and concerns.

  1. Abstract. Please check for language (this is important as readers will scan the Abstract first prior to further reading).

   We corrected the whole script with a help from a professional translator. 

  1. Paragraph 2.1. This section does well describe the subject selection procedures and includes a clear Figure on the process. But I suggest a line of text is attributed to how the authors dealt with the pitfall of selection bias? 

 In the process of collecting data (by interview), the target is confined to those who are healthy enough to conduct the interview. Participating in the National health and Nutrition Examination Survey means that they are healthy enough to respond to long-term surveys and measuring their body, and those with serious illness will be excluded from the subject. In our hypothesis, patients with severe hypertension and diabetes have increased RR values, and when those subjects are excluded, the RR approaches 1 (towards the null). Despite this limitation, RR greater than 1 in the analysis suggests that there is a clear correlation between hypertension and diabetic lung. If critically ill patients (who couldn't answer the interview) were included, the RR would be higher than now. So the selection bias does not affect the direction of our analysis results (which may affect the estimate value).

  1. Paragraph 2.4. Blood pressure is a major endpoint. Were blood pressure devices standardized and validated? I assume blood pressure was measured in the non-dominant upper arm according to RR?

  We inserted a detailed blood pressure measurement method. The inserted sentences are below.

'Blood pressure (BP) was measured by four nurses in charge of BP measurement in the special investigation department of KCDC. It was measured manually by a standard mercury sphygmomanometer (Baumanometer Wall Unit 33(0850), Baum
Co., Inc., Copiague, NY, USA) with a cuff of appropriate size. BP was measured three times (1 minute interval) in subjects in a sitting position after taking a rest at least 10 minutes, with the right arm supported at the level of the heart. The mean of the second and the third measurements was used in the analysis.'

  1. Paragraph 2.1. (...) How was the self-reported medical status quality controlled? (...) Paragraph 2.5. There are many measurements done in this study. How (and how often) were quality assurance procedures performed or was this a continuous process? Were there a priori set quality control requirements for the measurements?

  We inserted a detailed- quality control method. The inserted sentences are below.

 'KCDC and related academic societies have managed external and internal quality control programs for all steps (including survey administration, data collection, laboratory analysis and data processing). The laboratory data quality control program monitored laboratory performance to ensure all analytical values meet acceptable standards of precision and accuracy. In addition, 30 expert committees composed of over 120 experts have technically supported KNHANES regarding quality assurance and control of the survey and the selection of individual survey items.'

  1. Paragraph 2.6. Line 135. I am not an expert in the field of Korean liquors. What is Soju?

  We added a little explanation about Soju (with a bracket).

 'it corresponds to ≥ 7 glasses of Soju (Korean beverage which usually contains 20% of alcohol by volume) per occasion for men and ≥ 5 glasses of Soju for women.'

  1. A matter of semantics is the following (line 239): 'The clinical relevance for the individual might be small, but it cannot be neglected on the population level. Therefore, assessment of pulmonary function should be recommended for hyperglycemic population with poor blood pressure control'. This sentence may  be modified with the last part of the first sentence brought forward. The study warrants this approach in the conclusion: starting the sentence with 'The clinical relevance for the individual might be small,....etc.' is a too humble approach for the work done. (...) Line 298. The concluding sentences are phrased in a very careful and almost humble matter. In itself this is a very good approach from a philosphical perspective. I am not a reviewer that likes marketing blurps like 'Take home message' in scientific papers. However, clinicians tend to like some practical guidance. This Reviewer thinks the authors may, based on their well executed science, add a stronger clinical relevant statement in the concluding sentences to make the findings more palatable for the treating physicians, for example 'Our findings support the need for extensive clinical evaluation, treatment and follow-up in those presenting with diabetic lung and hypertension'.

We edited the content to show our result in a more confident manner. Thank you very much for your advice.

'As the numbers with diabetes and hypertension are ineluctable, our findings show a considerable clinical impact on population levels. Although the contribution of hyperglycemia and hypertension to total odds ratio is minor, in those diagnosed, clinical signs and symptoms do not tend to be isolated clinical findings. Therefore, extensive assessment of pulmonary function is warranted and should be recommended for hyperglycemic population with poor blood pressure control. An interesting fact, we found is that, when we designed this study, we planned to evaluate the effect of ‘prehypertension’ and ‘prediabetes’ because diabetes and hypertension develop gradually over many years and recent paradigm of medicine focuses on prevention rather than treatment. But the result showed us that prehypertension alone could not be regarded as a risk factor in the lung function deterioration.'

 Conclusion: 'We speculate that hypertension in combination with diabetes and not the diabetes itself might be associated with reduced lung function in the general adult population. Our findings support the need for pulmonary function evaluation, treatment and follow up on those presenting with diabetes and hypertension. However, it has to be taken into account that in our analyses the hypertension might just be an indicator for uncontrolled diabetes and that only the combination of very severe hypertension is associated with lung function impairment.'

We look forward to hearing from you in due time regarding our submission and to respond to any further questions and comments you may have. Once again thank you for your attentive review.

Sincerely,

J Lee, D Kwon, Y Lee, I Jung, D Hyun, H Lee and Y Ahn